# Effect of data quality improvement intervention on health management information system data accuracy: An interrupted time series analysis

Zewdie Mulissa[1]*, Naod Wendrad[2], Befikadu Bitewulign[1], Abera Biadgo[1], Mehiret Abate[1], Haregeweyni Alemu[1], Biruk Abate[3], Abiyou Kiflie[1], Hema Magge[1,4,5], Gareth Parry[6,7]

1 Institute for Healthcare Improvement, Addis Ababa, Ethiopia, 2 Medical Service Directorate, Federal Ministry of Health of Ethiopia, Addis Ababa, Ethiopia, 3 Policy, Planning, Monitoring and Evaluation Directorate, Federal Ministry of Health of Ethiopia, Addis Ababa, Ethiopia, 4 Division of Global Health Equity, Brigham and Women's Hospital, Boston, MA, United States of America, 5 Division of General Pediatrics, Boston Children's Hospital, Boston, MA, United States of America, 6 Institute for Healthcare Improvement, Boston, MA, United States of America, 7 Harvard Medical School, Longwood Avenue, Boston, MA, United States of America

* zmulissa@yahoo.com

**Data Availability Statement:** The data underlying the results presented in the study are available

## Abstract

### Background

As part of a partnership between the Institute for Healthcare Improvement and the Ethiopian Federal Ministry of Health, woreda-based quality improvement collaboratives took place between November 2016 and December 2017 aiming to accelerate reduction of maternal and neonatal mortality in Lemu Bilbilu, Tanqua Abergele and Duguna Fango woredas. Before starting the collaboratives, assessments found inaccuracies in core measures obtained from Health Management Information System reports.

### Methods and results

Building on the quality improvement collaborative design, data quality improvement activities were added and we used the World Health Organization review methodology to drive a verification factor for the core measures of number of pregnant women that received their first antenatal care visit, number of pregnant women that received antenatal care on at least four visits, number of pregnant women tested for syphilis and number of births attended by skilled health personnel. Impact of the data quality improvement was assessed using interrupted time series analysis. We found accurate data across all time periods for Tanqua Abergele. In Lemu Bilbilu and Duguna Fango, data quality improved for all core metrics over time. In Duguna Fango, the verification factor for number of pregnant women that received their first antenatal care visit improved from 0.794 (95%CI 0.753, 0.836; p<0.001) pre-intervention by 0.173 (95%CI 0.128, 0.219; p<0.001) during the collaborative; and the verification factor for number of pregnant women tested for syphilis improved from 0.472 (95%CI 0.390, 0.554; p<0.001) pre-intervention by 0.460 (95%CI 0.369, 0.552; p<0.001) during the collaborative. In

from Institute for HealthCare Improvement(IHI) http://www.ihi.org/.

**Funding:** The author(s) received no specific funding for this work.

**Competing interests:** The authors have declared that no competing interests exist.

Lemu Bilbilu, the verification factor for number of pregnant women receiving a fourth antenatal visit rose from 0.589 (95%CI 0.513, 0.664; p<0.001) at baseline by 0.358 (95%CI 0.258, 0.458; p<0.001) post-intervention; and skilled birth attendance rose from 0.917 (95%CI 0.869, 0.965) at baseline by 0.083 (95%CI 0.030, 0.136; p<0.001) during the collaborative.

## Conclusions

A Data quality improvement initiative embedded within woreda clinical improvement collaborative improved accuracy of data used to monitor maternal and newborn health services in Ethiopia.

## Introduction

Since October 2013, the Institute for Healthcare Improvement (IHI) has worked in partnership with the Ethiopian Federal Ministry of Health (FMoH), with the support of the Bill and Melinda Gates Foundation and Margaret A. Cargill Philanthropies to explore how quality improvement (QI) methodologies can accelerate progress of the FMoH to improve maternal and neonatal health in Ethiopia. As part of this work, woreda (woreda) based focused improvement collaboratives, based on the IHI Breakthrough Series collaborative [1] commenced in November 2016 with the aim of accelerating the progress in reducing maternal and neonatal mortality in three woredas. A key principle of the collaborative methodology is that participating teams submit and share data with each other, and, in particular, use a common core set of measures to understand changes in processes and outcomes over time. In the improvement collaboratives, a number of core process and outcome measures related to maternal and neonatal outcomes were taken from the national health management information system (HMIS) in Ethiopia. Additionally, the HMIS data was intended to be used as the basis for an overall summative impact evaluation of the improvement collaborative approach. However, numerous studies have reported problems with the quality of HMIS data, and many users do not trust these data [2–4].

Recent studies in Nigeria [5] and South Africa [6] reported improvement in data quality following quality improvement intervention as measured using a World Health Organization (WHO) data quality review methodology [3]. This WHO methodology can be employed from a facility level to a national level and provides an indication of the completeness of reporting and a verification factor to indicate data accuracy for specific reporting periods.

Prior to the start of the improvement collaboratives, HMIS data accuracy was assessed in the period May to October 2016, revealing inaccurate data in the core process and outcome metrics. We sought to avoid data falsification/intentional manipulation of data by focusing on improving data quality through training participants on the importance of high-quality data, monthly data review and feedback and trust building with health care leaders and workers. Consequently, the improvement collaboratives embedded a data quality improvement initiative aimed at improving the accuracy of maternal and neonatal health data from the HMIS system in participating sites. This paper describes the extent to which the accuracy of the HMIS data was improved during the improvement collaborative, as measured using the WHO review methodology.

## Materials and methods

### Improvement collaboratives

The Federal Ministry of Health (FMOH) of Ethiopia, regional health bureaus (RHBs) and IHI Ethiopia selected one woreda in each of the four most populous agrarian regions of Ethiopia to

introduce the improvement collaborative approach. The woredas were purposefully selected in consultation with the Federal Ministry of Health (FMoH) of Ethiopia and regional health bureaus (RHBs) based on pre-set criteria, including high maternal and perinatal deaths, high level of leadership commitment to improve the service and the absence of other partner organizations working on quality improvement project. All facilities in each woreda were included into a collaborative. This study includes results from the first three collaboratives introduced simultaneously at twenty health facilities in: Lemu Bilbilu (8), Tanqua Abergele (6) and Duguna Fango (6) woredas of Oromia, Tigray and Southern Nations and Nationalities People's region respectively.

The overall structure of the improvement collaboratives is summarized in Fig 1. In brief, the improvement collaboratives brought together teams from participating sites for 18 months to pursue a collective aim of improving maternal and child health care and outcomes across Ethiopia. Participating teams attended four 2 to 3-day learning sessions, where they came together to learn about the topic and to plan tests of change. During three Action Periods (time between learning sessions), teams were expected to use the Model for Improvement [7] to test changes using Plan-Do-Study-Act cycles in their local settings.

Each month the improvement collaboratives measured implementation progress using a number of core measures, including four sourced from HMIS data described in Table 1. We embedded additional activities focusing participating sites on improving the quality of selected

**Fig 1. Summary of the core improvement activities and embedded data quality improvement activities.**

**Table 1. HMIS-derived core measures used to track progress in the improvement collaboratives.**

| Core Measures Name | Definition | Data Source |
|---|---|---|
| Antenatal Care 1 | Number of pregnant women who had at least one antenatal care visit during their pregnancy. | National Health Management Information System (HMIS) |
| Antenatal Care 4 | Number of pregnant women who had four or more antenatal care visits during their pregnancy. | National Health Management Information System (HMIS) |
| Syphilis Screening | Number of pregnant women tested for syphilis | National Health Management Information System (HMIS) |
| Skilled Birth | Number of births attended by skilled health personnel | National Health Management Information System (HMIS) |
| Post-natal care 48 hours | Number of women who attended post-natal care at least once within 48 hours after delivery. | National Health Management Information System (HMIS) |

HMIS data associated with these specific measures. These activities were timed to coincide with the wider activities of the improvement collaboratives in Fig 1. We sought to avoid data falsification/intentional manipulation of data by focusing on improving data quality through training participants on the importance of high-quality data, monthly data review and feedback and trust building with health care leaders and workers. Training involved health care workers, health care leaders, the health care facility and woreda information officers who were responsible for data collection and was intended to help them understand that high-quality data are essential for improving service quality. Monthly data reviews and feedback were done by IHI Ethiopia senior project officers. Quarterly learning sessions and progress review meetings were used to reflect on progress of data quality among collaboratives and build trust with health care leaders and workers.

## Data accuracy measures

To understand how data accuracy changed from the pre-intervention period to intervention period to the post-Intervention period, we followed the WHO data review methodology [3] to create a measure of data accuracy for selected core measures as follows:

**Original HMIS report value.** Monthly, from May 2016 to December 2018, for each selected core measure, experienced IHI senior project officers collected data from the archive of HMIS reports at each facility.

**Recounted (audited) value.** Experienced IHI senior project officers undertook an audit by repeating the data collection, from May 2016 to December 2018, from standard antenatal care and delivery registers, developed by the Federal Ministry of Health (FMOH) of Ethiopia. This approach resulted in a monthly audited value for the selected core measures.

For each selected core measure, a monthly verification factor was calculated by dividing the recounted value by the original HMIS report value. A verification factor of 0.9–1.1 is considered "accurate"; <0.9 is considered over reported and a value > 1.1 as under reported [3].

## Data management and analysis

We used *Microsoft Excel 2016* for data entry and *STATA V13* for analysis. Mean and standard deviation were used for descriptive analysis. Interrupted time series was used to assess change in data accuracy, as measured using the verification factor, from the pre-intervention or baseline (May to October 2016) to during (November 2016 to December 2017) and post-intervention (January to December 2018) phases. Specifically, the time-series model for each measure, assessed whether a change in verification factor had occurred across each phase by adding a shift and slope-change term for each phase. Statistical significance was set at P < 0.05.

## Sample size

There are no fixed limits regarding the number of data points for interrupted time series study, as the power depends on various other factors including distribution of data points before and after the intervention, variability within the data, strength of effect and the presence of confounding effects such as seasonality [8]. For each facility, for each month, we used all data available from May 2016 to December 2018. We aggregated the data across facilities for each woreda, resulting in 32 data points for each woreda, 6 before and 26 after the start of the intervention.

## Ethical considerations

This research is part of a broader evaluation study that was reviewed and approved by Ethiopian Public Health Association (EPHA) Scientific and Ethical Review Committee. A letter of support was obtained from IHI Ethiopia project office.

# Results

## General Health facilities information

All health facilities in the three improvement collaboratives (20); Lemu Bilbilu, Tanqua Abergele and Duguna Fango provide antenatal care services including the first to fourth visits, Syphilis screening and skilled birth attendance (institutional delivery). Monthly HMIS reports and registers were also available and complete at all the health facilities in the improvement collaboratives; 3 primary hospitals and 17 health centers. All health facilities also reported their performances monthly to their respective woreda health offices.

Data personnel, HMIS officers, were available at all hospitals and 13 (76.5%) health centers. Training on data quality was given to all HMIS officers at hospitals and 3 (17.6%) health center staffs including HMIS officers. Data quality was checked monthly by lot quality assurance sampling (LQAS) method at all the hospitals and 15 (88.2%) of the health centers. As HMIS officers and relevant staff participated in the improvement collaboratives, they all received training on data quality.

## Verification factor

The mean & standard deviation of verification factors for core measure for each woreda over time period are summarized in Table 2. Post-natal care was frequently provided in other facilities, and within the scope of this study, it was not practical to undertake the data audit and the Post-natal care 48hours measure was dropped.

## Interrupted Time Series (ITS) analysis

For each time-series analysis, applying the Durbin-Watson test, suggested there was no auto-correlation in for all the verification factor measures. Additionally, we found no seasonality in the data. The results of the time series analysis are shown in Table 3 and illustrated in Fig 2. For each measure and Woreda, we show the coefficients from the best fitting model.

The verification factor for all four measures averaged 1 for all three time periods at Tanqua Abergele. Thus, we conducted the time series analysis only in Lemu Bilbilu and Duguna Fango.

For Antenatal Care 1 (Table 2 and Fig 2A), the interrupted time series for the verification factor for Lemu Bilbilu increased from 0.842 during the baseline period by 0.170 (95% CI 0.073, 0.267; p = 0.002) during the collaborative period and then fell by 0.116 (95% CI 0.194, 0.038; p = 0.007) during the follow-up period. For Duguna Fango, the verification factor rose

**Table 2. Verification factor by woreda for the baseline, intervention and post-intervention periods.**

| | Mean (standard deviation) | | |
|---|---|---|---|
| | **Baseline May 2016 to Oct 2016** | **Intervention Nov 2016 to Dec 2017** | **Post-Intervention Jan 2018 to Dec 2018** |
| **Antenatal Care 1** | | | |
| Lemu Bilbilu | 0.842 (0.123) | 1.012 (0.043) | 0.897 (0.133) |
| Duguna Fango | 0.794 (0.082) | 0.972 (0.039) | 0.963 (0.048) |
| Tanqua Abergele | 1 (0) | 0.983 (0.062) | 0.971 (0.099) |
| Overall | 0.817 (0.077) | 0.991 (0.018) | 0.923 (0.078) |
| **Antenatal Care 4** | | | |
| Lemu Bilbilu | 0.589 (0.131) | 0.866 (0.171) | 0.946 (0.147) |
| Duguna Fango | 0.486 (0.173) | 0.842 (0.148) | 0.982 (0.052) |
| Tanqua Abergele | 1 (0) | 0.990 (0.063) | 1.029 (0.069) |
| Overall | 0.549 (0.116) | 0.851 (0.142) | 0.964 (0.081) |
| **Syphilis Screening** | | | |
| Lemu Bilbilu | 0.664 (0.442) | 0.875 (0.235) | 1.058 (0.114) |
| Duguna Fango | 0.472 (0.141) | 0.933 (0.089) | 0.932 (0.101) |
| Tanqua Abergele | 1.045 (0.070) | 0.993 (0.014) | 0.982 (0.059) |
| Overall | 0.531 (0.195) | 0.899 (0.154) | 0.993 (0.067) |
| **Skilled Birth Attendance** | | | |
| Lemu Bilbilu | 0.917 (0.031) | 0.979 (0.049) | 1.024 (0.072) |
| Duguna Fango | 1.001 (0.040) | 0.961 (0.047) | 0.989 (0.019) |
| Tanqua Abergele | 1 (0) | 0.984 (0.078) | 1.008 (0.039) |
| Overall | 0.917 (0.031) | 0.975 (0.032) | 1.007 (0.032) |

from 0.794 during the baseline period by 0.173 (95% CI 0.128,0.219; p<0.001) during the collaborative period and no change in the follow-up period.

For Antenatal Care 4 (Table 2 and Fig 2B), the verification factor for Lemu Bilbilu increased from 0.589 during the baseline period by 0.358 (95%CI 0.258, 0.458; p<0.001) during the follow-up period. For Duguna Fango, the verification factor rose from 0.486 at baseline by 0.149 (95%CI 0.01, 0.287; p = 0.04) during the collaborative period and then by 0.349 (95%CI 0.223, 0.474; p<0.001) during the follow up period.

For Syphilis Screening (Table 2 and Fig 2C), in Lemu Bilbilu, the verification factor rose from 0.664 at the baseline by 0.296 (95%CI 0.064, 0.528; p = 0.018) during the collaborative period and no change in the follow-up period. In Duguna Fango the verification factor rose 0.472 at the baseline by 0.460 (0.369, 0.552; p<0.001) and no change in the follow-up period.

For Skilled Birth Attendance (Table 3 and Fig 2D), in Lemu Bilbilu, the verification factor rose from 0.917 at the baseline by 0.083 (95%CI 0.030, 0.136; p = 0.004) during the collaborative period and no change in the follow-up period. The relatively higher baseline verification factor for Duguna Fango (1.008) hasn't changed during collaborative and follow up periods. The overall finding is as shown in Fig 2 and Table 3.

## Discussion

This data quality improvement initiative, embedded within a wider set of improvement collaboratives, significantly improved accuracy of antenatal care visits and syphilis screening data within these improvement collaboratives in Ethiopia. The finding is comparable to a similar embedded data quality improvement initiative in South Africa [6] where data accuracy of public health facilities participating in an improvement initiative improved from 37% to 65%.

**Table 3. Results of the time series analysis for the verification factor for the core measures across woreda over the baseline, intervention and post-intervention phases.**

| | Baseline May 2016 to Oct 2016 | | | | Intervention Nov 2016 to Dec 2017 | | | | Post-Intervention Jan 2018 to Dec 2018 | | | |
| --- | --- | --- | --- | --- | --- | --- | --- | --- | --- | --- | --- | --- |
| | Constant | | Slope | | Constant | | Slope | | Constant | | Slope | |
| | Coefficient (95% CI) | P | Coefficient (95% CI) | P | Coefficient (95% CI) | P | Coefficient (95% CI) | P | Coefficient (95% CI) | P | Coefficient (95% CI) | P |
| **Antenatal Care 1** | | | | | | | | | | | | |
| Lemu Bilbilu | 0.842 | <0.001 | - | - | 0.170 | 0.002 | - | - | -0.116 | 0.01 | - | - |
| | (0.761, 0.923) | | | | (0.073, 0.267) | | | | (-0.194, -0.038) | | | |
| Duguna Fango | 0.794 | <0.001 | - | - | 0.173 | <0.001 | - | - | - | - | - | - |
| | (0.753, 0.836) | | | | (0.128, 0.219) | | | | | | | |
| **Antenatal Care 4** | | | | | | | | | | | | |
| Lemu Bilbilu | 0.589 | <0.001 | - | - | - | - | 0.037 | <0.001 | 0.358 | <0.001 | - | - |
| | (0.513, 0.664) | | | | | | (0.026, 0.048) | | (0.258, 0.458) | | | |
| Duguna Fango | 0.486 | <0.001 | - | - | 0.149 | 0.044 | 0.028 | <0.001 | 0.349 | <0.001 | - | - |
| | (0.405, 0.567) | | | | (0.01, 0.287) | | (0.015, 0.041) | | (0.223, 0.474) | | | |
| **Skilled birth attendance** | | | | | | | | | | | | |
| Lemu Bilbilu | 0.917 | <0.001 | - | - | 0.083 | 0.004 | - | - | - | - | - | - |
| | (0.869, 0.965) | | | | (0.03, 0.136) | | | | | | | |
| Duguna Fango | 1.008 | <0.001 | - | - | -0.034 | 0.064 | - | - | - | - | - | - |
| | (0.977, 1.039) | | | | (-0.069, 0.001) | | | | | | | |
| **Syphilis Screening** | | | | | | | | | | | | |
| Lemu Bilbilu | 0.664 | <0.001 | - | - | 0.296 | 0.018 | - | - | - | - | - | - |
| | (0.454, 0.873) | | | | (0.064, 0.528) | | | | | | | |
| Duguna Fango | 0.472 | <0.001 | - | - | 0.460 | <0.001 | - | - | - | - | - | - |
| | (0.39, 0.554) | | | | (0.369, 0.552) | | | | | | | |

CI = Confidence Interval, P = P-value.

Data quality improvement activities focusing on training, regular audits, monthly review and feedback have been successfully applied in other settings, without being embedded in a wider care improvement initiative. For example, in the United Republic of Tanzania [9] improved health information systems were associated with data use workshops actively engaging data users. In Kenya, on-site assessment and feedback was used to improve the accuracy of routine HIV health information [10]. In Peru, phone reminders to epidemiological surveillance teams and clinic visits were used to improve the timeliness and accuracy of data reported in an electronic surveillance system of infectious disease outbreaks. Phone reminders but not clinic visits improved timeliness, whereas in some settings data accuracy was improved by visits but not phone reminders [11]. Sequential data quality audits in six countries improved the quality and accuracy of data on immunization [12]. Elsewhere, improvement in data accuracy has been associated with increased use of real-time data entry using touch screen computers in Malawi [13], suggesting additional technology-driven approaches to improve data accuracy. Although a direct comparison is not possible, in all of these examples, the overall improvement

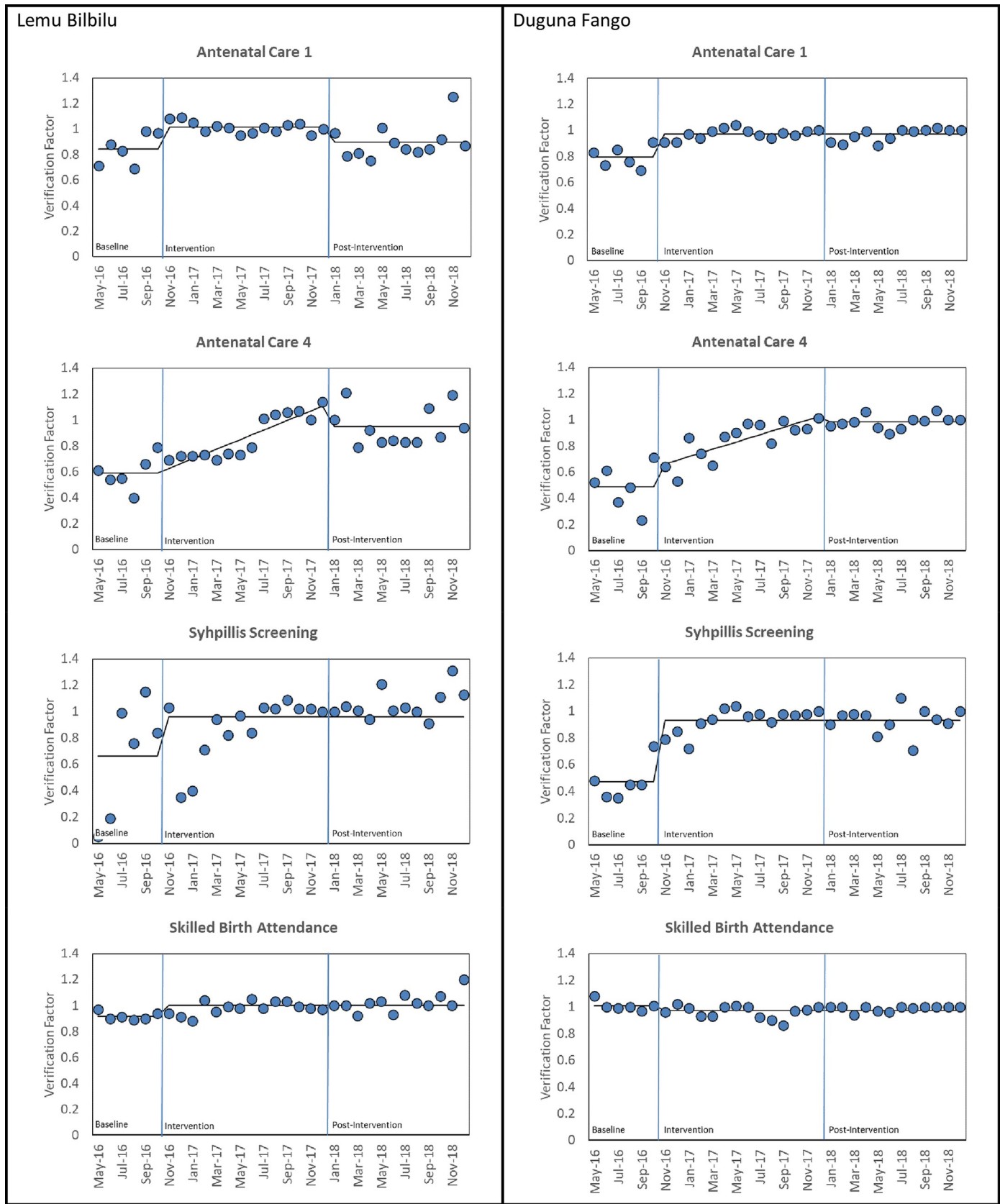

**Fig 2. Interrupted time series graph of change in verification factor from baseline to post-intervention.**

in data quality was less than in the embedded approach used in the current study in Ethiopia. This suggests that data quality improvement may be more successful when participants experience directly how the data is being used to provide a positive improvement in the care of their patients. Despite applying a similar approach to data accuracy, variation in improvement in data accuracy across collaboratives was reported. This may be due to variation in the issues faced within each collaborative, and shared learning within each collaborative not being similarly shared across collaboratives.

At baseline, a significant proportion of health facilities over reported syphilis screening (50%) and pregnant women that received antenatal care four visit (60%) as compared to pregnant women that received antenatal care first visit (35%) and Skilled Birth Attendance (15%). The finding is consistent with recent studies in south west [14] and southern [15] Ethiopia. This could be due to error in counting non-serial data elements from registers or intentional over reporting.

A limitation of this study is the use of data in a limited number of health facilities (20) and a lack of data on factors associated with data accuracy at baseline. The lack of 8 time points before intervention, limited the statistical power of the study [16]. Despite the improvement in data accuracy observed during this study, the absence of a comparison study arm limits our ability to conclude that there was a strong cause-and-effect relationship with the intervention. Part of the improvement may have resulted from the readiness to improve service quality. However, there have been no specific efforts to improve data quality in the study facilities or woredas at the time of the intervention.

Despite these limitations, the improvement in data quality observed in this study is encouraging, it suggests a similar approach of embedding data quality improvement efforts within a wider initiative where participants experience how the data can be used to improve care more broadly, could improve the quality of the data needed for decision-making and resource allocation in other public health programs.

## Conclusion

This study reports a simple, practical approach to improving the quality of public health information, both locally at health facility and in a woreda health information system. This data quality intervention improved both data accuracy on antenatal care1, antenatal care4 and syphilis screening in this study. Further research is needed to assess the effectiveness of similar data quality improvement approaches prospectively on a large scale.

## Acknowledgments

We would like to thank Institute for HealthCare Improvement (IHI), Ethiopia project office, Addis Ababa, Ethiopia for unreserved cooperation in providing data for this study and time for data analysis & manuscript writing.

## Author Contributions

**Conceptualization:** Zewdie Mulissa, Hema Magge, Gareth Parry.

**Data curation:** Zewdie Mulissa, Naod Wendrad.

**Formal analysis:** Zewdie Mulissa, Naod Wendrad.

**Investigation:** Befikadu Bitewulign, Abera Biadgo, Mehiret Abate, Haregeweyni Alemu.

**Methodology:** Zewdie Mulissa.

**Project administration:** Zewdie Mulissa, Befikadu Bitewulign, Abera Biadgo, Mehiret Abate, Haregeweyni Alemu, Abiyou Kiflie, Hema Magge.

**Resources:** Abiyou Kiflie, Hema Magge.

**Supervision:** Zewdie Mulissa, Hema Magge, Gareth Parry.

**Writing – original draft:** Zewdie Mulissa.

**Writing – review & editing:** Zewdie Mulissa, Naod Wendrad, Befikadu Bitewulign, Abera Biadgo, Mehiret Abate, Haregeweyni Alemu, Biruk Abate, Abiyou Kiflie, Hema Magge, Gareth Parry.

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
