## [Decision Letter · Decision Letter 0]

9 Jan 2020

PONE-D-19-32598

Effect of data quality improvement intervention on health management information system data accuracy: an interrupted time series analysis

PLOS ONE

Dear Mulissa,

Thank you for submitting your manuscript to PLOS ONE. After careful consideration, we feel that it has merit but does not fully meet PLOS ONE’s publication criteria as it currently stands. Therefore, we invite you to submit a revised version of the manuscript that addresses the points raised during the review process.

We would appreciate receiving your revised manuscript by 10 February 2020. To enhance the reproducibility of your results, we recommend that if applicable you deposit your laboratory protocols in protocols.io, where a protocol can be assigned its own identifier (DOI) such that it can be cited independently in the future. For instructions see: http://journals.plos.org/plosone/s/submission-guidelines#loc-laboratory-protocols

We look forward to receiving your revised manuscript.

Kind regards,

Russell Kabir, PhD

Academic Editor

PLOS ONE

Journal Requirements:

Reviewers' comments:

Reviewer's Responses to Questions

**Comments to the Author**

1. Is the manuscript technically sound, and do the data support the conclusions?

Reviewer #1: Partly

Reviewer #2: Partly

2. Has the statistical analysis been performed appropriately and rigorously? 

Reviewer #1: No

Reviewer #2: I Don't Know

3. Have the authors made all data underlying the findings in their manuscript fully available?

Reviewer #1: No

Reviewer #2: No

4. Is the manuscript presented in an intelligible fashion and written in standard English?

Reviewer #1: Yes

Reviewer #2: Yes

5. Review Comments to the Author

Reviewer #1: Thank you very much for giving an opportunity to review the present manuscript. This study sets out to describe the extent to which the accuracy of the HMIS data was improved during the improvement collaborative embedded data quality improvement initiative, as measured using the World Health Organization (WHO) data quality review methodology in Ethiopia. They found accurate data across all time periods in one region (Tanqua Abergele) while in two of the remaining three regions (Lem Bilbilu and Duguna Fango), data quality improved for all core metrics over time.

The manuscript needs some improvement in mainly the methods and results sections as a number of pages and lines should were not well described in the manuscript.

[Materials and methods]

1. Line 85 - The authors indicated that “one woreda in each of the four most populous agrarian regions of Ethiopia to introduce the improvement collaborative approach” but how the woreda were selected was not described.

2. Measures

Table 1: The authors definition for Antenatal Care 1 and Antenatal Care 4 were not that clear. I suggest the authors defined Antenatal Care 1 as “Number of pregnant women who had at least one antenatal care visit during their pregnancy”.

131 Data management and analysis

3. The authors did not mention any of the descriptive statistics used in this section.

4. What also informed the authors’ choice of median (inter-quartile range) instead of mean and standard deviation as the measure of central tendency and dispersion in the results section. If a test of normality was done, indicate the specific test statistic used.

5. 143-144: The authors described the selection of data used as consecutive sampling but go further to indicate that all data available during the study period was used. This is an indication of a census and not a sample as described by the authors.

6. What informed the authors choice of the slope change impact model used?

7. Did the authors assessed serial autocorrelation, non-stationarity and seasonality?

8. How was the fitness of the final selected model assessed? The authors should provide information on that.

9. Scatter plots of the various core measures over time should be added to help visualize the distribution of the data.

10. What was the level of significance used?

Results

11. 157-158 The authors mentioned nothing about what happened to the health centres without data personels and the 82.4% of health center staffs who were not trained. If nothing was done for them, if nothing was done for them won’t it affect the expected results?

12. An overall verification factor for the whole 3 regions will be important to be added in Table 2.

13. Verification factor for the intervention and post intervention period combined will be very informative since the impact of the intervention is not expected to take longer to be released.

14. Overall time series analysis for the verification factor for the core measures for the whole 2 regions combined will be important to be added in Table 3.

15. Time series analysis for the verification factor for the core measures for the intervention and post intervention periods combined will be very informative since the impact of the intervention is not expected to take longer to be released

16. Foot notes should be added to Tables to explain abbreviations used (CI, P).

17. The interpretations for the result should preside the tables.

18. The interpretation of Table 3 results was poorly done . it was explained in only line 178

19. Nothing was mentioned on the significance of the changes observed.

Conclusion

20. The conclusion should be rewritten to be based on the findings of the study.

Reviewer #2: This paper is may be interesting but very difficult to understanding for reader. Actually I can not understand the objectives of this paper. even i think the findings of this paper may be not sound. i think advanced statistical analysis may be upgrade the quality of this paper.

6. PLOS authors have the option to publish the peer review history of their article (what does this mean?). If published, this will include your full peer review and any attached files.

Reviewer #1: No

Reviewer #2: No

---

## [Author Response · Author response to Decision Letter 0]

13 Jul 2020

Reviewer#1

[Materials & Methods]

1. Line 85 – The authors indicated that “one woreda in each of the four most populous agrarian regions of Ethiopia to introduce the improvement collaborative approach” but how the woreda were selected was not described.

We have added details of the Woreda selection (line 88).

2. Measures

Table1: The authors definition for antenatal care1 and antenatal care4 were not that clear. I suggest that authors defined antenatal care1 as “Number of pregnant women who had at least one antenatal care visit during their pregnancy”.

Thank you, we have redefined Antenatal care1 and antenatal care4 as per the suggestion.

131: Data management and analysis

3. The authors didn’t mention any of the descriptive statistics used in this section

We have added details of the descriptive statistics used in the data management and analysis section (line 138).

4. What also informed the authors’ choice of median (inter-quartile range) instead of mean and standard deviation as the measure of central tendency and dispersion in the result section. If a test of normality was done, indicate the specific test statistic used.

As we used mean for the time series analysis, we replaced all median (inter-quartile range) throughout the paper with mean (standard deviation).

5. 143-144: The authors described the selection of data used as consecutive sampling but go further to indicate that all data available during the study period was used. This is an indication of a census and not a sample as described by the authors.

We have clarified this description in the Sample Size section.

6. What informed the authors choice of the slope change impact model used?

The study referred to in the introduction, set in South Africa by Mphatswe W et al (Bull World Health Organ. 2012) indicated data accuracy changed shortly after the data quality intervention. This led us to hypothesize that a similar finding may occur in the current study, and thus we applied a model that would allow us to detect changes in the slope and overall average (step change) after the introduction of the data quality improvement efforts in the current study.

7. Did the authors assessed serial autocorrelation, non-stationarity and seasonality?

Yes, we checked autocorrelation using Durbin-Watson d-statistic and it is zero (0) suggesting the outcomes are independent. We found no non-stationarity and seasonality. We have added details of this into the results section (line 179).

8. How was the fitness of the final selected model assessed? The authors should provide information on that.

We have added details to the results section describing how we used standard approaches to assess fitness, using residual plots and examination of autocorrelation and seasonality, as described in #7 above.

9. Scatter plots of the various core measures over time should be added to help visualize the distribution of the data.

We have updated the Figure 2, to show both the individual data points and the fitted models. 

10. What was the level of significance used?

We used a significance level of 0.05 and have added this to the methods section.

Results

11. 157-158 The authors mentioned nothing about what happened to the health centers without data personnels and the 82.4% of health center staffs who were not trained. If nothing was done for them, won’t it affect the expected results?

We have clarified this section in the text. The data quality improvement activities were embedded within a wider improvement collaborative. Part of the underlying theory of an improvement collaborative is that attendees will take their learning back to their home facilities and apply it with local staff. 

12. An overall verification factor for the whole 3regions will be important to be added in Table2.

We have added the overall verification factor to Table 2

13. Verification factor for the intervention and post intervention period combined will be very informative since the impact of the intervention is not expected to take longer to be released.

The data quality improvement initiative was embedded within a wider quality improvement collaborative. As such the data quality improvement activities were introduced and built upon over time and ended when the wider improvement collaborative was completed. The post intervention period is thus a different period to the intervention period allowing us to assess sustainability. Consequently, we believe it is important to keep the intervention and post-intervention periods separate.

14. Overall time series analysis for the verification factor for the core measures for the whole 2regions combined will be important to be added in Table3.

Quality improvement initiatives are reliant on local engagement with activities, and on local context. This leads to the timing of improvement occurring, or not, to often differ from one setting to another. Consequently, we choose to focus of the time series analysis on the two Woredas separately, rather than combine them, which can risk to important variation being lost. As such, we argue strongly not to combine the results across the two Woredas, and to focus on learning from the individual woredas.

15. Time series analysis for the verification factor for the core measures for the intervention and post intervention periods combined will be very informative since the impact of the intervention is not expected to take longer to be released.

As described in our response to #13 above, we believe it is important to keep the intervention and post-intervention phases separate.

16. Foot notes should be added to tables to explain abbreviations used (CI, P)

We have added the foot notes as suggested.

17. The interpretations for the result should preside the tables.

We have re-positioned the tables so the description of the results comes before them.

18. The interpretation of table3 results was poorly done. It was explained in only line 178.

We have updated the wording to this section to better describe the analysis.

19. Nothing was mentioned on the significance of the changes observed.

We have included reference to the relevant p-values in the results section.

Conclusion

20. The conclusion should be rewritten to be based on the findings of the study.

We have re-written the conclusion to be more closely based on the findings of the study (line 258). 

Reviewer#2: This paper may be interesting but very difficult to understand for the reader. Actually, I can’t understand the objectives of this paper, even I think the findings of this paper may be not sound. I think advanced statistical analysis may upgrade the quality of this paper.

Based on the responses to Reviewer #1, we believe this paper is now clearer, and is statistically sound.

---

## [Editor Report · Decision Letter 1]

3 Aug 2020

Effect of data quality improvement intervention on health management information system data accuracy: An interrupted time series analysis

PONE-D-19-32598R1

Dear Dr. Mulissa,

We’re pleased to inform you that your manuscript has been judged scientifically suitable for publication and will be formally accepted for publication once it meets all outstanding technical requirements.

Kind regards,

Russell Kabir, PhD

Academic Editor

PLOS ONE
---

## [Editor Report · Acceptance letter]

5 Aug 2020

PONE-D-19-32598R1 

Effect of data quality improvement intervention on health management information system data accuracy: An interrupted time series analysis. 

Dear Dr. Mulissa:

I'm pleased to inform you that your manuscript has been deemed suitable for publication in PLOS ONE. Congratulations! Your manuscript is now with our production department. 

Kind regards, 

on behalf of

Dr. Russell Kabir 

Academic Editor

PLOS ONE